# Implementation of a Full Momentum Conservative Approach in Modelling Flow Through Tidal Structures

**Nejc Čož** , **Reza Ahmadian \*** and **Roger A. Falconer**

Hydro-environmental Research Centre, School of Engineering, Cardiff University, Cardiff CF24 3AA, UK; CozN@cardiff.ac.uk (N.Č.); FalconerRA@cardiff.ac.uk (R.A.F.)

\* Correspondence: AhmadianR@cardiff.ac.uk; Tel.: +44-29-2087-4003

**Abstract:** Understanding the impact of various hydraulic structures, such as coastal reservoirs and tidal range impoundments, has been one of the key challenges of hydro–environmental engineering in recent years. Over the last half-century, several proposals for tidal range schemes in the UK have been scrutinised and then abandoned due to the uncertainty over the environmental footprint and/or the cost of electricity. Therefore, it is essential to understand the fundamental assumptions for reliable hydrodynamic analysis of these projects. This study examined the impact of the fully conserved momentum through tidal structures using a novel approach. The method was applied to 2D and 3D versions of the regional model of Swansea Bay tidal lagoon, examining two different types of velocity distribution at turbine exit. A simplified distribution significantly increased the velocity and length of the jet exiting the turbines during power generation. A realistic distribution gave more accurate results, with jet velocities more closely resembling the situation without including the momentum. The 3D model with realistic distribution has markedly improved the resulting vertical velocity profile. The value of the improved methodology for momentum conservation has proved to be particularly useful in local-scale studies. It can be applied to other similar hydraulic structures and used for the analysis of sediment transport, water quality, etc.

**Keywords:** hydraulic structures; numerical modelling; momentum conservation; tidal range; Delft3D; tidal modelling

## 1. Introduction

Numerical models have been widely used in recent decades in designing and evaluating the environmental impacts of hydraulic structures, such as coastal reservoirs and tidal impoundments. It is crucial to represent such hydraulic structures accurately in numerical models, particularly since this type of modelling plays a key role in the design process and impact assessment of such structures [1,2]. Renewable energy is currently one of the most rapidly growing sectors in the global and UK's domestic power industry [3]. In recent years, we have seen a significant increase in research of renewable technologies, including tidal energy. The concept of Tidal Range Structures (TRS) dates back to the 1960s, when the first tidal range barrage was constructed in La Rance, France [4]. Since then only a small number of TRSs have been developed around the world, with the uncertainty around the hydro–environmental impacts of such schemes being one of the key reasons for a lack of development of these schemes. Accurate representation of tidal range schemes in hydro–environmental models, for the reasons presented below, is one of the key considerations of this study. However, the methodology outlined herein can be implemented to many other hydraulic structures such as coastal reservoirs and large-scale flood barriers.

Tidal range power plants take advantage of the potential energy stored in the tidal amplitude. The fundamental approach is to create an enclosed basin by constructing a tidal barrage or lagoon, fitted with turbines and sluice gates, which can be controlled to create an appreciable head difference, which is built up either side of the impoundment wall. Once the head difference reaches an optimum level, water is allowed to flow through the turbines installed in the barrier, which then generates electricity. This can be achieved both on incoming and outgoing tides, and with two tidal cycles per day; this results in four periods of power generation each day [5].

The west coast of the UK has one of the largest potential regions for tidal power generation in the world [6]. The region of Severn Estuary and Bristol Channel regularly experiences tidal amplitudes in the range of over 10 m [7]. This makes them highly suitable for tidal range energy extraction. The region is well known for its exceptional tides, and with the proximity to the UK's power grid, this represents an excellent site for the world's first sizeable TRS.

There have been several studies on various aspects of TRSs. A number of them have applied 2D numerical models to investigate the operation and environmental impacts of different TRS schemes. Xia et al. [8,9], Kadiri et al. [10], Ahmadian [11,12], and Athanasios et al. [2,13] have investigated the hydro–environmental impacts of a proposed Severn barrage in the Severn Estuary. In all cases, the models have provided useful predictions for a variety of designs and operations, including: investigation of the regional hydrodynamics, sediment transport and water quality, optimisation of power generation efficiency and coastal flood risk assessment. More recently, there have been several studies undertaken on modelling tidal lagoons in the UK, including studies: on Swansea Bay lagoon [2,13–15], North Wales lagoons [16] and Cardiff lagoon [13,17] to name but a few. Adcock et al. [18] discussed the open boundary problem that can occur due to the operation of large TRSs, such as barrages in an estuary. The impacts on the flow conditions can be potentiality so far-reaching that they can have a noticeable effect on the model's open boundaries. This requires expansion of the computational domain further into the ocean, beyond the usual open boundary. Continental shelf models have been employed by Bourban et al. [19] in a comprehensive study on the potential of renewable energy extraction in the UK. On the other hand, sometimes simplified numerical models can prove more effective than expensive 2D models. For example, Aggidis et al. [20] and Xue et al. [15] both showed that 0-D models are useful tools for assessment of power production and are commonly used for operational optimisation of TRSs. However, there are certain aspects of TRSs, such as bed erosion and turbine performance, that require a much more detailed approach to numerical modelling [21]. For example, an experimental study by Jeffcoate et al. [22,23] employed a high-resolution CFD model for assessment of shear stress on the bed downstream of a tidal barrage. Similarly, a study by Wilhelm et al. [24] simulated flow through a bulb turbine using high-resolution RANS and Large Eddy Simulation models for assessment of the flow field through a TRS turbine.

The importance of accurately simulating the hydrodynamic impact of hydraulic structures while modelling TRSs have been highlighted in previous studies by Ahmadian et al. [25], Bray et al. [1] and Angeloudis et al. [16]. One of the key aspects of simulating hydraulic structures is calculating accurately the volume of water transferred through the structures with the appropriate speed, namely conserving mass and momentum. Although, conservation of mass has been studied in detail in the past, very little work has been done on conserving momentum through the structure. A more accurate understanding of the local flow complexity can have a significant impact on bed erosion and sediment deposition [10,26], which were beyond the focus of this study. In addition, a realistic turbine representation gives a more informed view on the efficiency of power production [2] and provides a better estimation of the local flow velocity, which is also important for the safety assessment on local shipping due to the TRS's proximity to shipping lanes.

In this study, a new model has been developed that includes momentum conservation for the discharge through hydraulic structures, such as turbines and sluice gates. The method was applied to the operation of the proposed Swansea Bay lagoon in the Bristol Channel. We have investigated the importance of the correct approach to momentum treatment in the context of the hydrodynamic

modelling for TRSs. We were interested in the extent of the impact on the regional hydrodynamics, as well as the local flow patterns around the TRS. The simulations were run both in 2D and 3D configurations to investigate the extent of the three-dimensional flow conditions that develop near the turbine housing on both sides of the TRS.

## 2. Swansea Bay Tidal Lagoon

Tidal lagoons are conceptually a novel adaptation of an existing and proven technology of tidal barrages [27]. Instead of spanning the entire width of an estuary, the lagoon is created by constructing an embankment in a horseshoe shape attached to the coastline. In the same way as a barrage, the lagoon takes advantage of the large tidal amplitude to create a head difference across the structure that drives flow through the turbines. Swansea Bay is potentially a suitable site for deployment of such a scheme due to its location on the north bank of the Bristol Chanel which, together with the Severn Estuary, represents one of the largest tidal range resources in Europe [6].

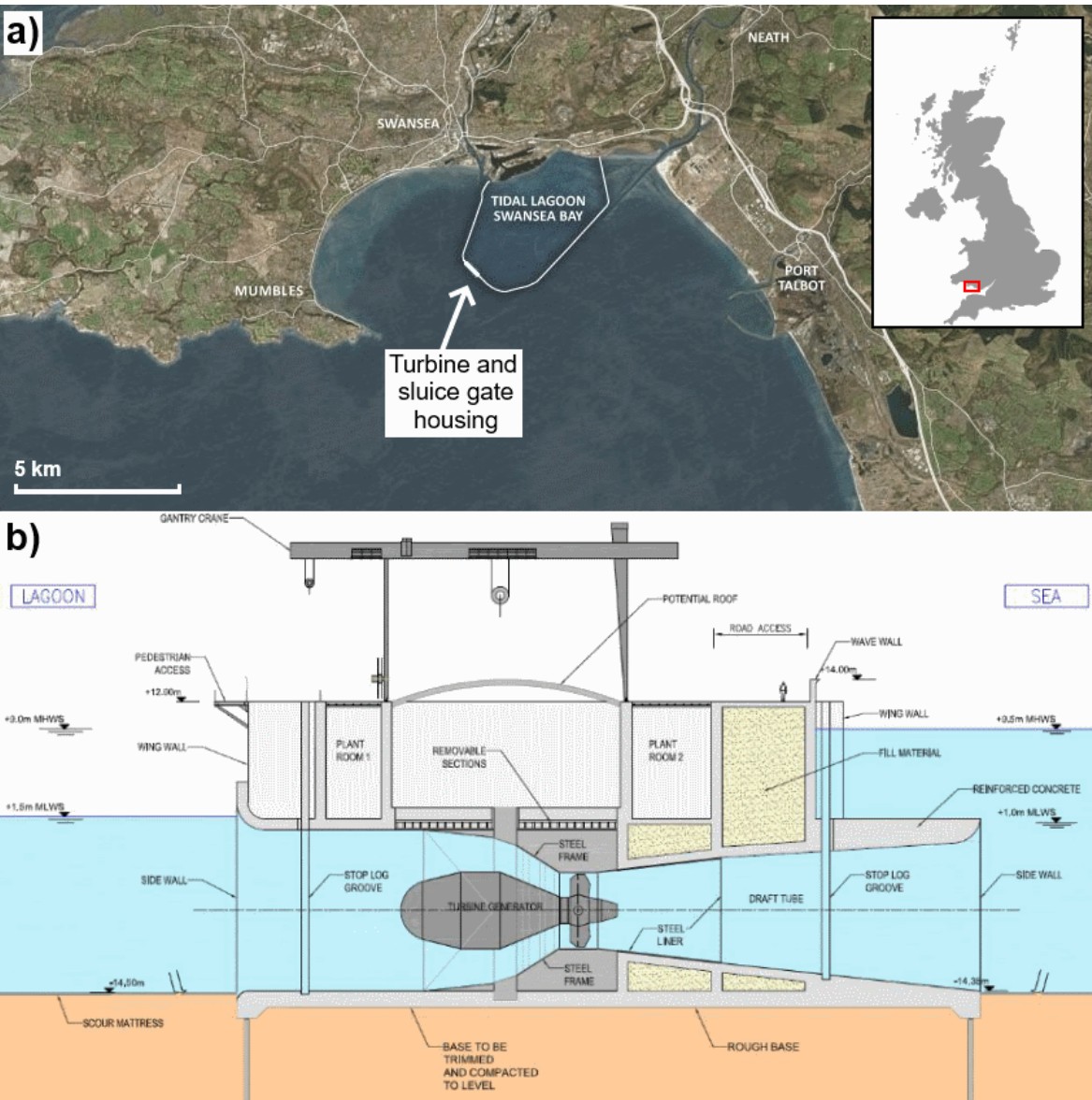

**Figure 1.** Location of the current lagoon proposal within Swansea Bay (**a**) and a cross-section schematic of the housing section of a bulb turbine, designed specifically for the Swansea Bay lagoon (**b**). Images courtesy of Tidal Lagoon Plc.

A tidal lagoon in Swansea Bay was first proposed more than a decade ago. In 2015, the project received a development consent order (DCO) from the UK government and in 2016 a backing from an independent review, namely the Hendry Review of Tidal Lagoons, commissioned by the UK government [28]. Since the initiation of the project, several alternative proposals with different lagoon layouts and turbine capacities were considered. The shape and layout of the structure used in this study was taken from the latest proposal put forward by Tidal Lagoon Plc [29] and is shown in Figure 1a.

If built, Swansea Bay tidal lagoon would become the largest TRS to-date and the first of its kind in the world. The scheme proposes a 9.5 km long embankment that would form an impoundment, with a surface area of 11.5 km$^2$ [30]. The proposed power plant would have an installed capacity of 320 MW, providing power to over 155,000 homes [28]. This would be achieved through a set of sixteen bulb turbines (Figure 1b), each with a diameter of 7.2 m and an output power of 20 MW, with an additional 800 m$^2$ of sluice gates to aide with the maximisation of the head difference. The location of the turbine housing, where the turbines and sluice gates are located, is shown in Figure 2. This design allows for a two-way operation, where power is produced during both ebb and flood tides. The starting and minimum head for generation were designed to be 2.5 m and 1.5 m respectively. The same characteristics were also used in other recent studies, such as Ref. [2,13,31].

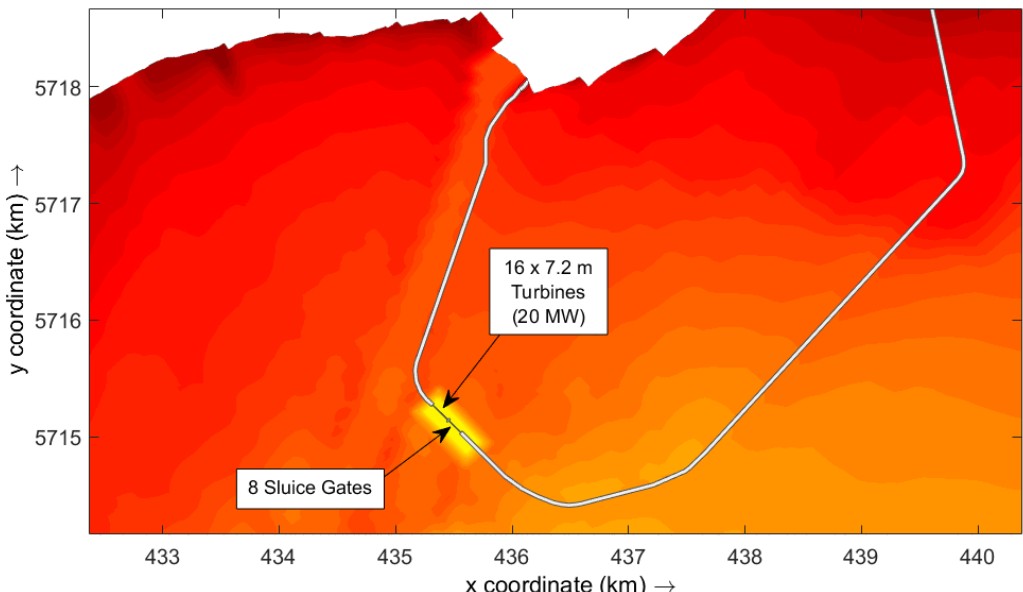

**Figure 2.** Close-up of the lagoon, showing the location of the turbines and sluice gates in the numerical model.

## 3. Methodology

### 3.1. Delft3D and Governing Equations

The software used for hydrodynamic simulations was Delft3D-FLOW (referred to as Delft3D from here on), which is an open source code developed by Delatares in the Netherlands. Delft3D is capable of solving both 2D (depth-averaged) and 3D unsteady flow problems, resulting from tidal and meteorological forcing, and using a curvilinear grid [32]. The continuity Equation (1) and Navier–Stokes Equations (2) and (3) are solved for incompressible free surface flow using the hydrostatic pressure assumption and are respectively given as:

$$\frac{\partial \zeta}{\partial t} + \frac{1}{\sqrt{G_{\xi\xi}}\sqrt{G_{\eta\eta}}} \frac{\partial \left( (d+\zeta)u\sqrt{G_{\xi\xi}} \right)}{\partial \xi} + \frac{1}{\sqrt{G_{\xi\xi}}\sqrt{G_{\eta\eta}}} \frac{\partial \left( (d+\zeta)v\sqrt{G_{\eta\eta}} \right)}{\partial \eta} = (d+\zeta)(q_{in} - q_{out}) \quad (1)$$

$$\frac{\partial u}{\partial t} + \frac{u}{\sqrt{G_{\xi\xi}}}\frac{\partial u}{\partial \xi} + \frac{v}{\sqrt{G_{\eta\eta}}}\frac{\partial u}{\partial \eta} + \frac{\omega}{(d+\zeta)}\frac{\partial u}{\partial \sigma} - \frac{v^2}{\sqrt{G_{\xi\xi}}\sqrt{G_{\eta\eta}}}\frac{\partial\sqrt{G_{\eta\eta}}}{\partial \xi} + \frac{uv}{\sqrt{G_{\xi\xi}}\sqrt{G_{\eta\eta}}}\frac{\partial\sqrt{G_{\xi\xi}}}{\partial \eta}$$

$$- f_v = -\frac{1}{\rho_0 \sqrt{G_{\xi\xi}}}P_\xi + F_\xi + \frac{1}{(d+\zeta)^2}\frac{\partial}{\partial \sigma}\left(\nu_V \frac{\partial u}{\partial \sigma}\right) + M_\xi \quad (2)$$

$$\frac{\partial v}{\partial t} + \frac{u}{\sqrt{G_{\xi\xi}}}\frac{\partial v}{\partial \xi} + \frac{v}{\sqrt{G_{\eta\eta}}}\frac{\partial v}{\partial \eta} + \frac{\omega}{(d+\zeta)}\frac{\partial v}{\partial \sigma} + \frac{uv}{\sqrt{G_{\xi\xi}}\sqrt{G_{\eta\eta}}}\frac{\partial\sqrt{G_{\eta\eta}}}{\partial \xi} - \frac{u^2}{\sqrt{G_{\xi\xi}}\sqrt{G_{\eta\eta}}}\frac{\partial\sqrt{G_{\xi\xi}}}{\partial \eta}$$

$$+ f_u = -\frac{1}{\rho_0 \sqrt{G_{\eta\eta}}}P_\eta + F_\eta + \frac{1}{(d+\zeta)^2}\frac{\partial}{\partial \sigma}\left(\nu_V \frac{\partial v}{\partial \sigma}\right) + M_\eta \quad (3)$$

where $u$ and $v$ are the flow velocities in the $\xi$ and $\eta$ directions for a horizontal curvilinear coordinate system respectively; $\zeta$ is the water level above a reference datum, while $d$ is the depth below the reference datum and together they form the total depth $H$; $\omega$ is the vertical velocity related to the moving $\sigma$ plane; $f_u$ and $f_v$ are the Coriolis parameters; $\rho_0$ is the density of water; $\nu_V$ is the vertical eddy viscosity; $P_\xi$ and $P_\eta$ are the pressure gradients; forces $F_\xi$ and $F_\eta$ represent the imbalance of the horizontal Reynold's stresses; $q_{in}$ and $q_{out}$ are the local source and sink terms per unit volume ($s^{-1}$), $M_\xi$ and $M_\eta$ represent the contributions due to external body forces ($m/s^2$), such as sources or sinks of momentum resulting from hydraulic structures; and $(\sqrt{G_{\xi\xi}})$ and $(\sqrt{G_{\eta\eta}})$ are the transformation coefficients between the curvilinear grid and rectangular coordinate system.

In the horizontal plane, the domain is defined with an orthogonal curvilinear grid. In the vertical direction, a $\sigma$-coordinate system is used, consisting of one or more layers bounded by $\sigma$-planes. The layers themselves are not strictly horizontal but are fitted proportionally between the bottom and free surface. The vertical velocity of the moving layers $\omega$ is computed from the continuity Equation (1) and can be associated with the upwelling and downwelling motions of the layers. The physical vertical velocity w is not involved in the model equations and is calculated during post-processing if required. For 2D simulations, only one computational layer is selected, and the problem translates to solving the depth-averaged equations. Delft3D is based on the finite difference numerical method for democratization of the differential equations, and the state variables are arranged into a staggered grid. An Alternating Direction Implicit (ADI) method is used for time integration to deliver second order accuracy in space.

## 3.2. Setup of Hydrodynamic Model

The tidal flows in Swansea Bay have been extensively studied in Refs. [2,8,11,33,34]. The hydrodynamics in the region have been shown to be highly complex due to the exceptional tidal range, the complex land boundary and the highly variable gradients of the seabed elevations [8]. However, the estuary was found to be well-mixed with no evidence of stratification, suggesting that natural flow conditions are primarily two-dimensional [34]. A high-resolution 2D shallow water model is considered suitable for such a study. In order to fully model the lagoon, this study employed a regional scale model covering the entire Severn Estuary and Bristol Channel, that is presented in Figure 3. Such large extent of the model was selected to ensure an appropriate location for the boundaries, enabling the hydrodynamic processes within the area of interest to be captured.

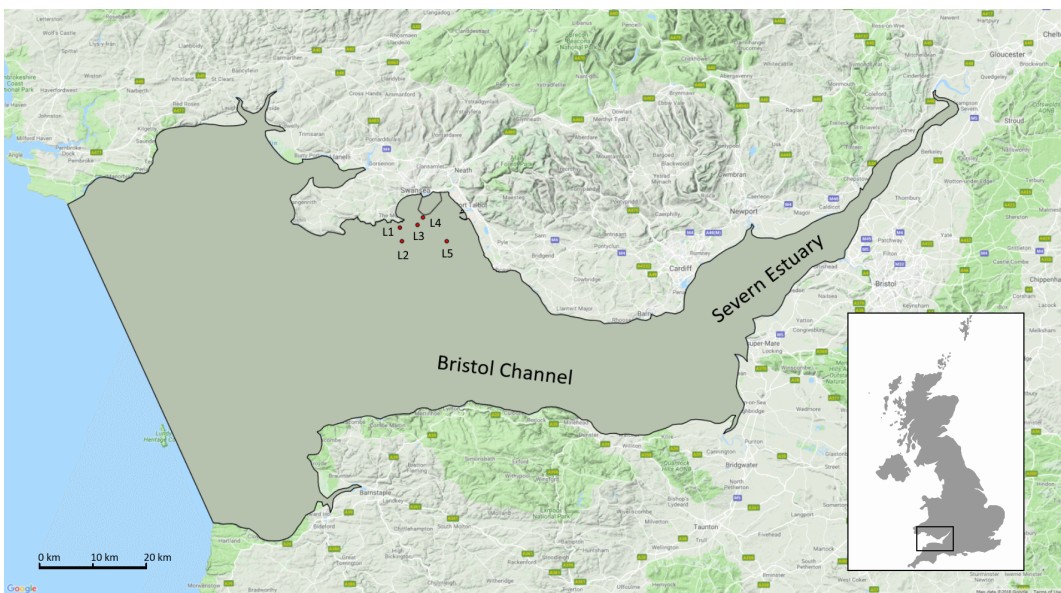

**Figure 3.** Computational domain for the assessment, with the outline of the proposed tidal lagoon in Swansea Bay and with red dots showing the locations of validation sites (background map from Google Maps service).

A rectangular, curvilinear grid with a variable mesh resolution was employed to increase the efficiency by reducing the mesh size only in the areas of interest. The grid resolution in and around the lagoon was therefore relatively high at 30 m × 30 m, while retaining a coarser mesh size of 500 m × 500 m at the edge of the domain. For simplicity, the Manning's n was set to a constant value of 0.03 across the entire domain. The model could be further improved by calibration with more complex formulations of the roughness coefficient related to the bed characteristics. However, this was considered beyond the immediate scope of this study given that the focus has been on demonstrating the relative impacts of the momentum conservation method on the jets and wakes exiting the turbines and with the general bed characteristics across the computational domain being fairly uniform and constituting sand and generally smooth bed slope and morphological changes. It was therefore assumed that spatial changes in the Manning's roughness coefficient (without appropriate data) would have little impact on the main thrust of this study, which was to assess the impact of including improved momentum conservation characteristics as potential energy is converted to kinetic energy through the turbines and sluice gates. The seaward boundary was located at the entrance to the Bristol Chanel, which stretched from Heartland Point in south-west England, to the south-west tip of Pembrokeshire in Wales. This boundary was set up as a water level boundary, with a time series of tidal amplitude. Hourly data were obtained from the National Oceanography Centre Continental Shelf Model [35]. The boundary condition time-series covered a full neap-spring tidal cycle between 19 January and 2 February 2012 in 1 h intervals (Figure 4). Bathymetry was generated from the Seazone data, relative to Chart Datum (CD), at a 30 m grid resolution [36,37]. The data was first converted to Mean Sea Level (MSL) and then interpolated onto the model's grid. The bathymetry of the model is shown in Figure 5.

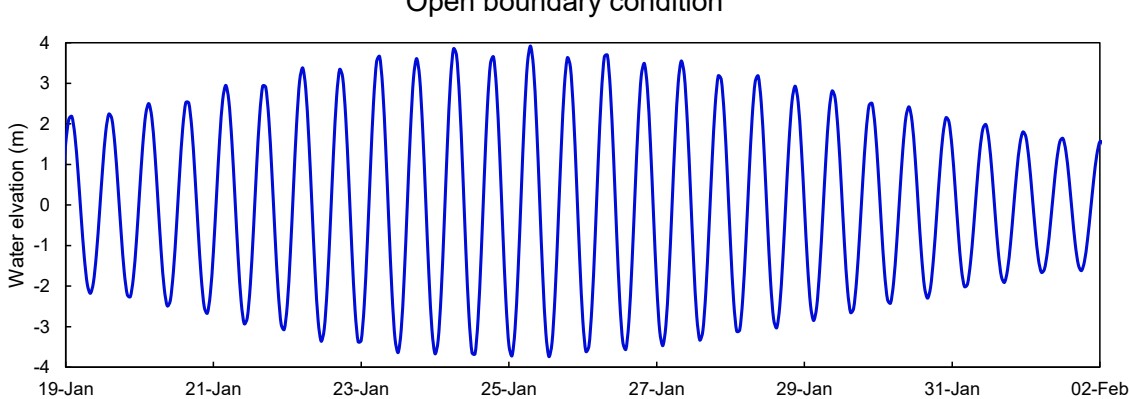

**Figure 4.** Water elevation time-series at the open boundary covering a full neap-spring tidal cycle.

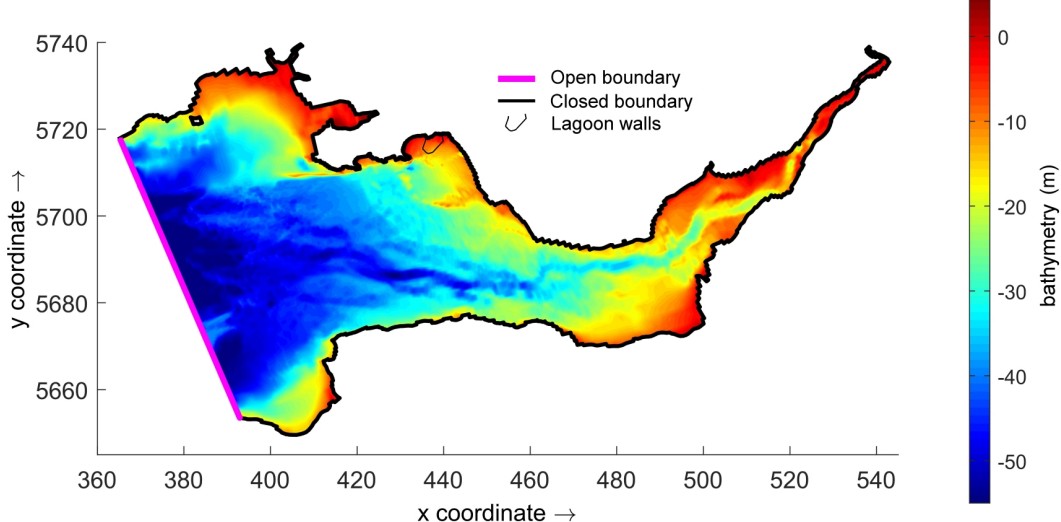

**Figure 5.** Bathymetry of the Severn Estuary and Bristol Channel obtained from EDINA Digimap service.

### 3.3. Model Validation

Validation of the hydrodynamics was carried out for the 2D depth-averaged model without the lagoon in place. The results were compared with the measured data across the domain and particularly in Swansea Bay, where the tidal lagoon is proposed to be built and therefore is expected to have the most impact. Observation sites L1 to L5, pictured in Figure 3, were selected for comparison due to their location in and around the main area of interest in terms of assessing the hydro–environmental changes, i.e. in the vicinity of the lagoon. The measured data was provided through the Smart Coasts project, an EU-funded research study on water quality of bathing waters in Wales. An Acoustic Doppler current profiler (ADCP) was deployed at five different locations in the bay (Figure 3). Pressure sensor was used for recording the water elevation by converting the raw pressure data to height above the sensor using density values calculated using a pre-set water column, mean salinity value and measured temperatures. The current profile was recorded every 10 minutes over an averaging period of 60 s. Only the tidal stream component of the observed velocity was used, Ignoring the mean drift and inconsistent residual drift components of the current, o, which was depth-averaged for comparison of the 2D model. The validation was preformed over a 14-day period to capture the typical neap-spring tidal variation.

Figures 6 and 7 show the validation results for sites L3 and L5 respectively. Each figure shows the results for three different state variables, namely water elevation, depth-averaged velocity magnitude and its direction relative to true North. The results showed good agreement with the observed data.

They were also consistent with the results reported in the literature, where different numerical models, such as DIVAST and EFDC, have been used in the past to predict the of hydrodynamic conditions in the Severn Estuary and Bristol Channel [1,11,38,39].

To quantify the performance of the hydrodynamic model the Nash-Sutcliffe model efficiency (NSE) was calculated for the state variables using the following Equation (4):

$$NSE = 1 - \frac{\sum_{i=1}^{n}(O_i - S_i)^2}{\sum_{i=1}^{n}(O_i - \overline{O})^2} \tag{4}$$

where $O_i$ are the observed data points, $S_i$ are the simulation results, and $\overline{O}$ is the mean of the observed data. An efficiency value of 1 corresponds to an exact match between predictions and observations, while a value of 0 indicates that the model predictions are as accurate as the mean of the observed data and becoming less accurate if the value falls below zero. NSE was initially developed to assess the forecasting performance of hydrological models. However, it can be used for evaluation of any predictive model when comparing it to observed data [40]. Analysis of predicted state variables for all five observation sites are presented in Table 1.

**Table 1.** NSE efficiency of model predictions at five different locations shown in Figure 3.

| Location | Water Elevation | Velocity Magnitude | Current Direction |
|---|---|---|---|
| L1 | 0.982 | 0.776 | 0.840 |
| L2 | 0.987 | 0.953 | 0.922 |
| L3 | 0.984 | 0.928 | 0.874 |
| L4 | 0.985 | 0.898 | 0.791 |
| L5 | 0.988 | 0.967 | 0.878 |

The NSE values for water elevation ranged from 0.982 to 0.987 and suggested a high level of agreement between the model results and observations. Analysis of the current velocity magnitudes produced NSE values of 0.9 and higher, except at the site L1 where the NSE value is slightly lower at 0.776. However, this lower value is still within an acceptable range. Similarly, values calculated for the current directions were between 0.791 and 0.922, again showing strong agreement between the two sets of results. The slightly lower values for the latter variable were due to the sensitivity of the NSE method to data outliers. As observed in Figures 6 and 7, both the predicted and observed data for velocity direction contained "spikes" which occurred during slack water and when the velocity falls close to zero. At the same time, the flow must change the direction between an incoming and outgoing tide, which appears as outlier points in the flow direction time series. The direct comparison of the results in Figures 6 and 7, together with the quantitative NSE analysis, indicated a robust forecasting ability of the model and justified its use for further hydrodynamic analysis with TRSs in place.

*3.4. Tidal Lagoon Modelling*

The numerical model of the tidal lagoon was created by first splitting the existing computational domain into two disconnected subdomains, namely the area impounded by the lagoon and the rest of the domain. The divide between the two domains prevents any flow from passing from one side to the other. Computational cells on the opposite sides of the barrier, where turbines and sluice gates are located, were dynamically linked by a prescribed discharge relationship, which is a function of the water level difference across the barrier. The operation of the TRS was simulated by opening and closing the turbines and sluice gates, based on pre-programmed operation sequences. This was achieved by transfer of mass between the computational cells that represented intake and outfall of hydraulic structures. The transfer of mass is included in the continuity Equation (1) through local sink and source terms.

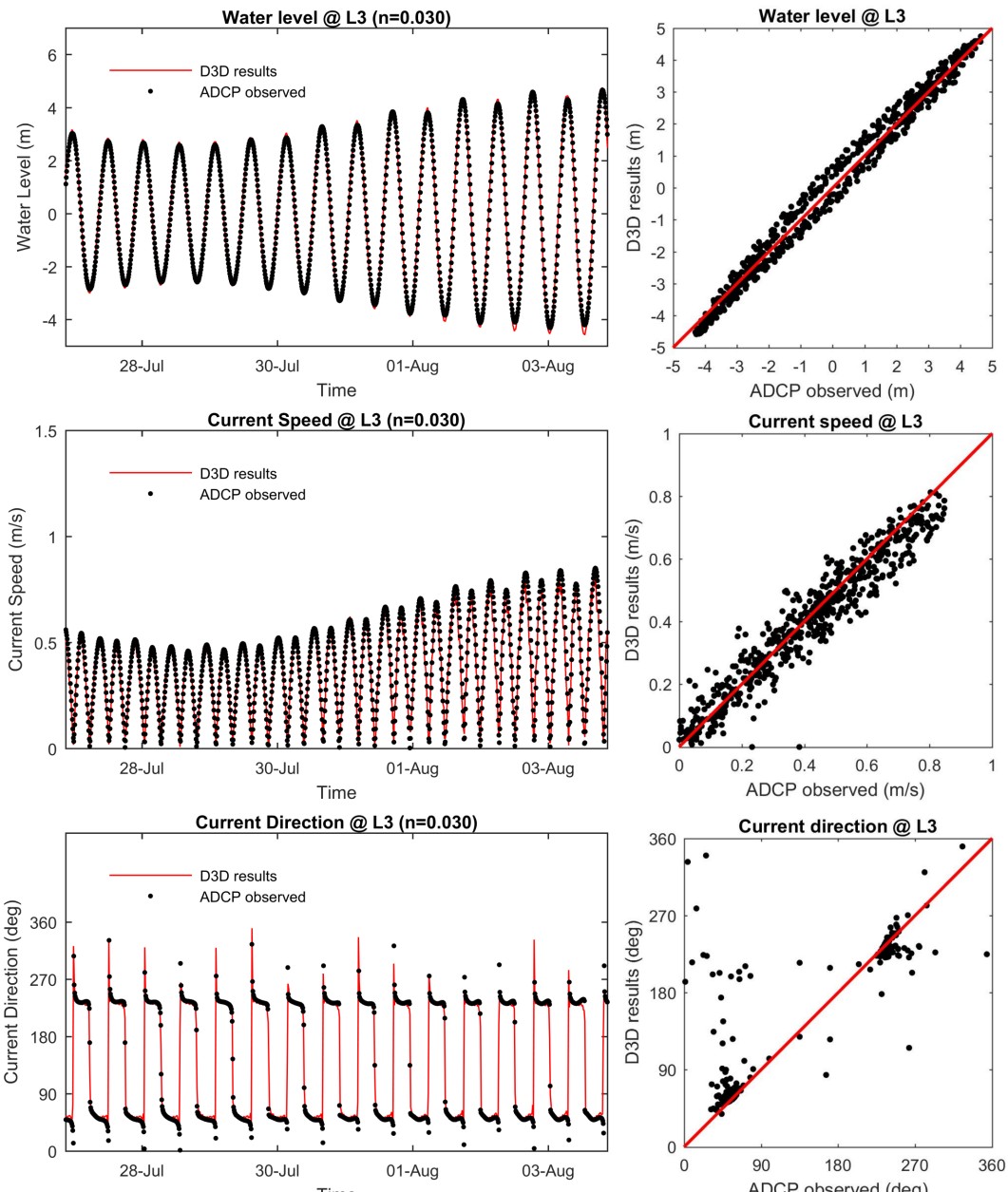

**Figure 6.** Validation results at location L3, showing time series of water levels and flow velocity (**left**) and scatter plots comparing the observations and simulation results (**right**).

The operation of the lagoon can be optimised to generate the maximum energy [15]. Since this is outside the scope of this study, only a typical two-way generation scheme (as suggested by Tidal Lagoon Plc [29]) was considered herein. A two-way generation scheme can generally be split into six distinct phases, as illustrated in Figure 8. A holding stage is first initiated as soon as the water elevation inside the lagoon reaches its peak. During this stage, both turbines and sluice gates are closed, retaining the water level inside the lagoon at the highest possible elevation, while the water level outside is receding on the ebb tide. Once a sufficient head difference has been established, the turbines are opened and start generating as the ebb-generation phase begins. During generation the water level on both sides begins falling until the head difference is reduced to the minimum efficient head for power production. At that point, the sluice gates are opened and the turbines are disconnected from the generator (but still allow water to flow through them). This phase is called a sluicing stage, where water leaves the lagoon to obtain the lowest possible elevation inside the impoundment, before both

hydraulic structures are closed again for the second holding phase. With the flood tide now underway, the whole procedure is repeated with the water level rising outside the lagoon and generating power by filling the lagoon, until it reaches the peak water elevation and before the next cycle begins.

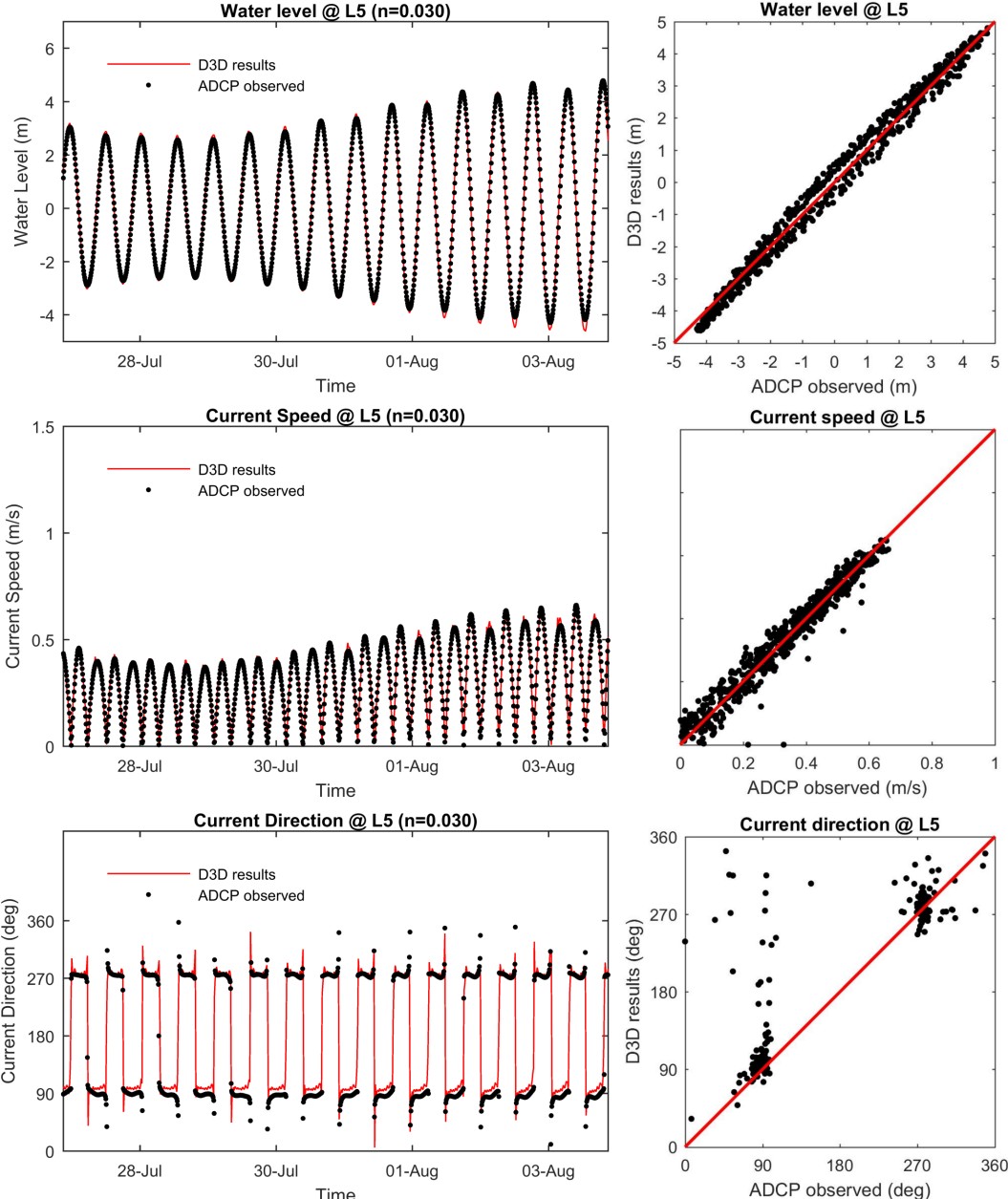

**Figure 7.** Validation results at location L5, showing time series of water levels and flow velocity (**left**) and scatter plots comparing the observations and simulation results (**right**).

The instantaneous discharge that was passed between the two domains during the generation and sluicing stages was determined by the water level difference on either side of the lagoon. For sluice gates the flow was calculated using the orifice Equation (5):

$$Q_S = C_d A_S \sqrt{2gH} \tag{5}$$

where $Q_S$ is the discharge through sluice gate in m$^3$/s; $C_d$ the discharge coefficient; $A_S$ the area of the sluice gate in m$^2$, $g$ is gravitational acceleration; and $H$ is the water level difference across the lagoon wall.

The instantaneous discharge through the turbine $Q_T$ was obtained from a predefined Q-H relationship between the discharge and head difference, namely the hill chart [31]:

$$Q_T = f(H) \tag{6}$$

A hill chart is unique for every type of turbine and is generally obtained experimentally by the manufacturing company. In the absence of a specific hill-chart being provided for this scheme, a typical hill chart for the Andritz Hydro double-regulated bulb turbine, previously used for tidal lagoons by Baker [41], was implemented herein. The resulting Q-H relationship for a 7.2 m diameter turbine is shown in Figure 9.

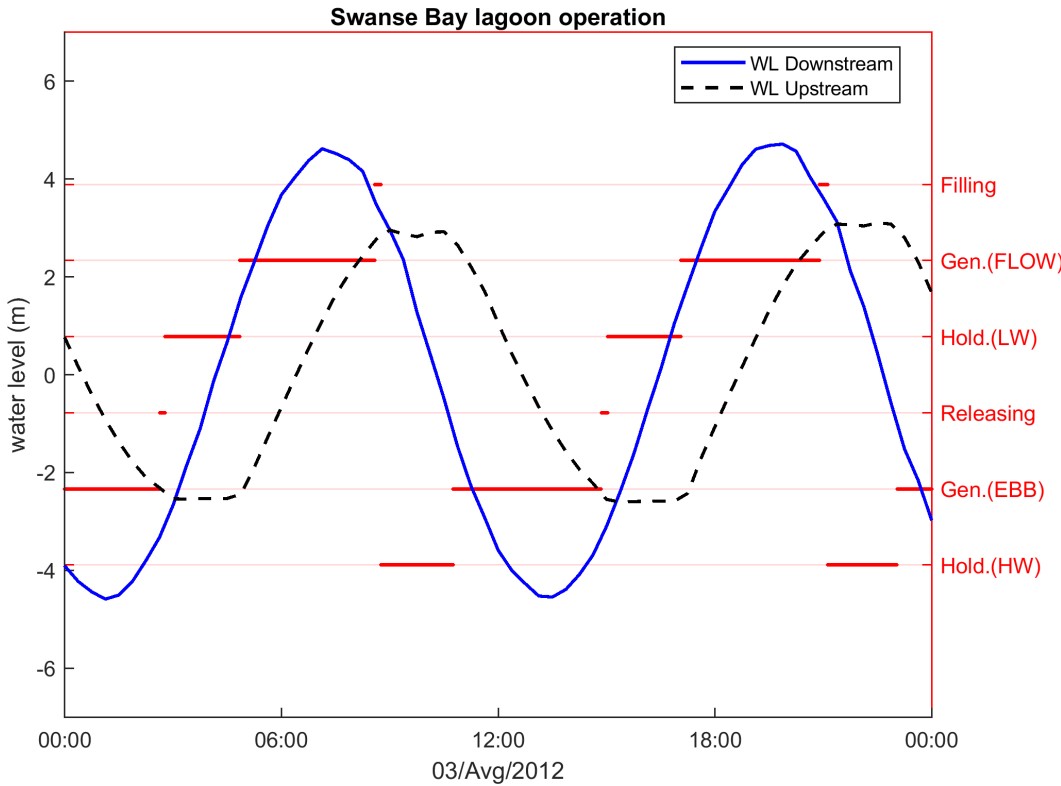

**Figure 8.** Two-way generation sequence represented by water elevation on either side of the lagoon.

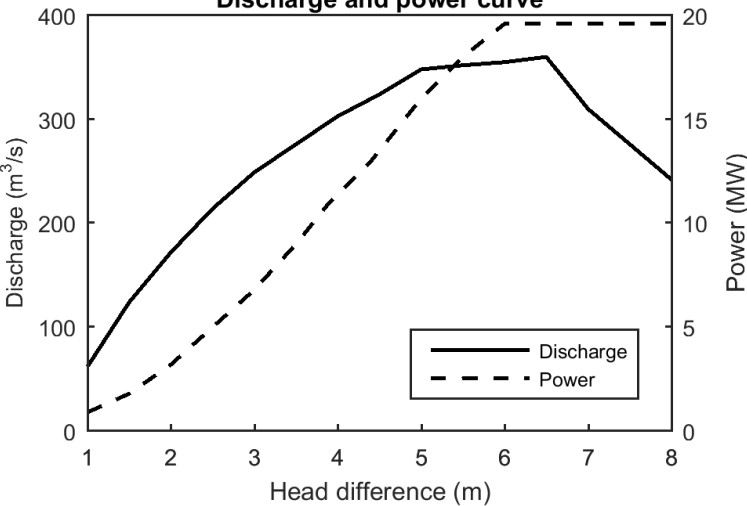

**Figure 9.** Q-H and Power-H relationship characteristic for bi-directional turbines used in the study.

In addition to a Q-H relationship we can also determine a power-head (P-H) relationship for the turbine by employing the Equation (7):

$$P = \eta(H)\rho g H Q_T \tag{7}$$

where $\eta$ is an overall efficiency for the turbine in relation to the available head difference, and $\rho$ is the density of the fluid.

*3.5. Momentum Conservation*

Angeloudis et al. [2] highlighted that the discharge velocity exiting the turbines can easily exceed 10 m/s and therefore has a significant influence on the nearby velocity field. This high velocity can significantly impact the local water quality, sediment transport, and other morphological and ecological processes, both inside and outside the basin. Therefore, the velocity of jets exiting the turbines must not be neglected, even within larger regional scale hydrodynamic models. The local 3D effects in the vicinity of the turbines also affects the surface elevation profile, as illustrated by Jeffcoate et al. [22] in their experiments. The depth-averaged model inaccurately predicted the water depth in the region of 20 turbine diameters downstream of the lagoon, while the results of the 3D model showed much better agreement with the measured data. Since the operation of the lagoon is governed by the difference in water elevation on both sides of the turbine, an inaccuracy in water level predictions may lead to an erroneous turbine flowrate and power output.

There are different conceptual approaches to model the correct transfer of momentum through tidal range turbines. The most accurate method would be a full 3D computational fluid dynamics (CFD) model, where the flow through a turbine itself would be included in the computations [42]. However, such simulations are computationally very demanding and too expensive to be used at the regional scale. A more conventional method is the technique of domain decomposition, where the two subdomains (one upstream of the structure and another downstream) are physically separated and dynamically linked at the location of the turbines. Instead of modelling a complex flow through the turbines on a sub-grid scale, only the information of relating to the time varying flow rate is passed on through the link. The treatment of this unique boundary condition is therefore paramount to the accuracy of the flow conditions in the area of the turbine exit. The approach used by Angeloudis et al. [2] was to artificially change the water depth across the boundary to a value that forces the cell interface to be of an equivalent area as the actual turbine cross-section.

In this study a different approach was used, where the momentum was conserved by adding an extra term into the momentum equations. The term represents an external force that is acting on the body of water due to the high-velocity jet entering the domain [43]. An equivalent concept is widely used in tidal stream power, where the stream turbine acts as a momentum sink as it takes the energy out of the passing water flow [44]. The same approach in modelling tidal stream turbines, where momentum sink was represented by an external force equal and opposite to the thrust force exerted by the flow on the propeller, has already been introduced to Delft3D by several authors in Ref. [45,46]. While the approach of momentum sink within the computational domain is already commonly used in tidal stream studies, the momentum source represents a novel method of modelling the TRSs and other hydraulic structures. In Delft3D this momentum source term is denoted as $M_\xi$ and $M_\eta$ in Equations (2) and (3). The force of a jet entering a body of water can be characterised as the rate of change of momentum of the fluid and is commonly defined by Equation (8):

$$\vec{F}_T = \rho_0 Q(\vec{v}_1 - \vec{v}_2) \tag{8}$$

where $\rho$ is water density, $Q$ is the flow rate of the jet entering the control volume cross-section, $\vec{v}_1$ and $\vec{v}_2$ are velocities of the jet entering and exiting the control volume, respectively. For the sake of simplicity Equation (8) can be rewritten in scalar form. From here on the derivation will be shown only for a component in the $\xi$ direction (9). However the same is applied to the component in $\eta$ direction:

$$F_T = \rho_0 Q(\hat{u} - u) \tag{9}$$

where $\hat{u}$ is the velocity of the incoming jet and u the background velocity in the target cell. Assuming the force is acting on the whole volume of the computational cell and taking into account Newton's second law of motion, a body force is divided by the mass of water in the computational cell $m_{cell}$. The momentum source terms $M_\xi$ and $M_\eta$ in Equations (2) and (3) can now be rewritten into the following form (10):

$$M_\xi = \frac{F_{T\xi}}{m_{cell}} = \frac{F_{T\xi}}{\rho_0 V_{cell}} \tag{10}$$

where $V_{cell}$ is the volume of the targeted computational cell in m$^3$. Substituting Equation (9) into Equation (10) gives (11):

$$M_\xi = \frac{\rho_0 Q(\hat{u} - u)}{\rho_0 V_{cell}} = \frac{Q}{V_{cell}}(\hat{u} - u) \tag{11}$$

and we can simplify it to (12):

$$M_\xi = q_{in}(\hat{u} - u) \tag{12}$$

where $u$ is obtained from the existing velocity field, while $\hat{u}$ and $q_{in}$ are calculated from the turbine (5) or sluice gate (6) discharge during the power generation and sluicing phases of the lagoon's operation (13) and (14):

$$\hat{u} = \frac{Q_T}{A_T} \tag{13}$$

$$q_{in} = \frac{Q_T}{V_{cell}} \tag{14}$$

where $A_T$ is the flow-through area of the turbine and $V_{cell}$ the volume of the computational cell.

*3.6. Representation of Turbine Discharge*

A high-resolution 2D modelling approach is commonly considered suitable for a regional scale study of tidal energy projects in estuaries [2]. Due to the modifications to the momentum treatment of the turbine discharge, an increase in the flow complexity was expected in the vicinity of the lagoon that would be unlikely to be captured using conventional depth-averaged models. Therefore, an additional 3D model with five horizontal layers was also constructed for this study. The vertical dimension was modelled by splitting the domain proportionally into five layers, where each layer was still considered depth-averaged. The horizontal velocities of adjacent layers were coupled by the vertical advection and the vertical viscosity term [32]. For the calculation of the vertical turbulent velocity and eddy diffusivity, a *k-ε* turbulence model was used, which included a second-order turbulence closure model [47]. The new 3D model retained the same resolution of the horizontal grid from the original 2D model as well as all other attributes, such as the boundary conditions, bathymetry, bottom roughness etc. Five different model revisions, including different representations of the turbine, were implemented for this study, as summarised in Table 2. Firstly, a 2D model simulation with no modifications to the momentum treatment (SBL1) was set up to act as a baseline for the subsequent model set-ups. The other four scenarios included the implementation of two different velocity distributions for the exiti ng jet, that were applied to both 2D and 3D models. These velocity distribution scenarios are summarised below and are shown in Figure 10.

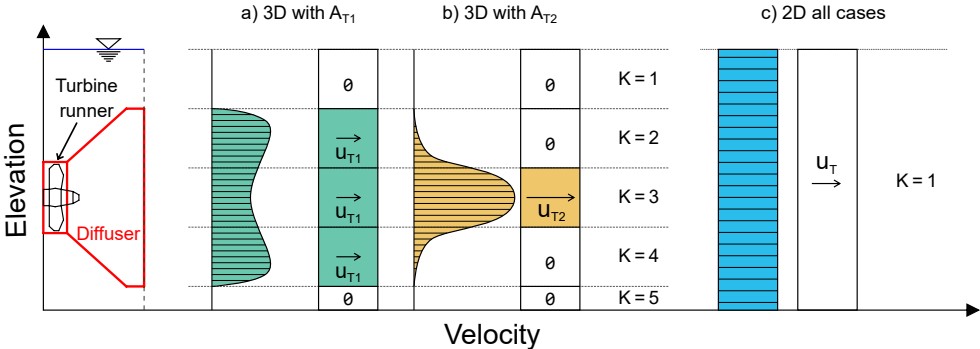

**Figure 10.** Representation of velocity distribution in the numerical model for a 3D and 2D configuration.

### 3.6.1. Realistic Velocity Distribution

As illustrated in Figure 11, horizontal bulb turbines are designed with a draft tube in a shape of a diffuser behind the turbine runner. The role of the draft tube is to minimise head losses and increase the efficiency of the turbine, by increasing the static pressure while simultaneously decreasing the velocity of the exiting flow [48]. The shape of the velocity profile was acquired from the experimental work by Wilhelm et al. [24] and was distributed over the square-shaped area of the diffuser. For the case of the 5-layer model, this velocity was equally divided amongst the middle three layers as shown in Figure 10a. The velocity for momentum conservation in Equation (13) was therefore calculated using the square-shaped area at the diffuser exit ($A_{T1}$) as shown in Figure 11. This velocity distribution has been applied to scenarios SBL2 and SBL4 (Table 2).

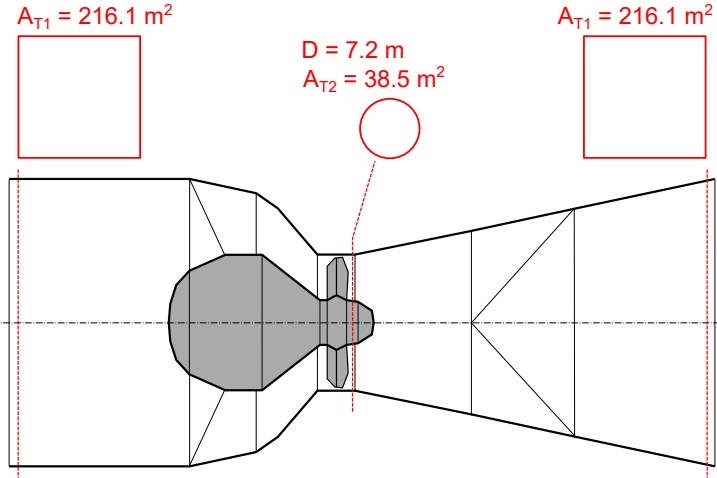

**Figure 11.** Turbine cross-section - flow through area at the duct entrance and exit is rectangular and it transitions to a smaller circular shape in the middle at the location of the turbine runner.

### 3.6.2. Simplified Velocity Distribution

In this case, the velocity is concentrated in a narrow jet, assuming the shape of a bell curve as illustrated in Figure 10b. The discharge was applied to a single horizontal layer that was located at the same location as the vertical position of the turbine's centreline. The energy loss due to the expansion of the flow through the diffusor was ignored. The velocity for momentum conservation in Equation (13) was calculated using the turbine flow-through area at the runner ($A_{T2}$) as shown in Figure 11. This velocity distribution has been applied to scenarios SBL3 and SBL5 (Table 2).

**Table 2.** Summary of the five different scenarios investigated in this study.

| Dimension | | 2D | | 3D | |
|---|---|---|---|---|---|
| Simulation Run | SBL1 | SBL2 | SBL3 | SBL4 | SBL5 |
| Momentum conservation | No | Yes | Yes | Yes | Yes |
| Velocity distribution | - | Realistic | Simplified | Realistic | Simplified |
| $A_T$ | | $A_{T1}$ | $A_{T2}$ | $A_{T1}$ | $A_{T2}$ |
| (m$^2$) | - | 216.1 | 38.5 | 216.1 | 38.5 |

## 4. Results

### 4.1. Velocity Field

Figures 12 and 13 show the instantaneous velocity fields near the lagoon during the flood and ebb generation phases respectively. Velocities from the 3D models have been depth-averaged to allow for a direct comparison with the 2D model. Changes to the velocity pattern were limited to the vicinity of the lagoon and have never extended further than 2.5 km away. The changes were induced in the form of a high-velocity jet that has alternatingly formed on the inside and outside of the lagoon. The figures correspond to the peak flow rates during a spring tidal cycle, where the velocity magnitude is at its extreme. However, the extreme velocities are only an example that represents a general trend which was observed throughout the simulation period, regardless of the size of the tidal range. The peak velocities at the turbine exit for each scenario are collected in Table 3.

**Table 3.** Peak flow velocity ($u_t$) and instantaneous discharge ($Q_T$) just downstream of the turbine exit for a typical spring tidal cycle during flood and ebb power generation, and the corresponding head difference ($H_T$). Velocities for cases SBL4 and SBL5 are presented both with a depth-averaged value ($u_T$) and with separate values for respective computational layers ($u_{Tk}$).

| Scenario | Layer | $u_T$ (m/s) | $u_{Tk}$ (m/s) | $Q_T$ (m$^3$/s) | $H_T$ (m) | $u_T$ (m/s) | $u_{Tk}$ (m/s) | $Q_T$ (m$^3$/s) | $H_T$ (m) |
|---|---|---|---|---|---|---|---|---|---|
| | | | | **Ebb** | | | | **Flood** | |
| SBL1 | K = 0 | 1.37 | - | 4408 | 3.49 | 1.32 | - | 4336 | 3.47 |
| ine SBL2 | K = 0 | 1.48 | - | 4411 | 3.50 | 1.41 | - | 4377 | 3.48 |
| ine SBL3 | K = 0 | 2.05 | - | 5364 | 4.60 | 1.97 | - | 5192 | 4.48 |
| ine | K = 1 | | 1.06 | | | | 0.99 | | |
| | K = 2 | | 1.50 | | | | 1.20 | | |
| SBL4 | K = 3 | 1.47 | 1.40 | 4381 | 3.47 | 1.40 | 1.27 | 4348 | 3.44 |
| | K = 4 | | 1.23 | | | | 1.09 | | |
| | K = 5 | | 0.93 | | | | 0.62 | | |
| ine | K = 1 | | 0.88 | | | | 1.05 | | |
| | K = 2 | | 1.89 | | | | 1.95 | | |
| SBL5 | K = 3 | 1.76 | 2.75 | 4973 | 4.13 | 1.84 | 2.77 | 4954 | 4.17 |
| | K = 4 | | 2.18 | | | | 2.19 | | |
| | K = 5 | | 1.90 | | | | 1.90 | | |

The flow patterns in scenarios SBL2 and SBL4 with a realistic velocity distribution (Figures 12b,d and 13b,d) were very close to the baseline model SBL1 (Figures 12a and 13a), with a slight increase in the velocity at the turbine exit. For example, during the ebb generation, the velocity increased from 1.37 m/s in SBL1 to 1.48 m/s in SBL2 and 1.47 m/s in SBL4. The average increase in velocity was under 10 per cent for both the SBL2 and SBL4 scenarios. As expected, scenarios SBL3 and SBL5, with a simplified velocity distribution, saw a more significant increase in the velocity coming out of the turbines (Figures 12c,e and 13c,e). In the 2D model (scenario SBL3), the velocities increased by almost 50 per cent. For example, the peak velocity during ebb tide increased from 1.37 to 2.05 m/s and the peak velocity during flood tide increased from 1.32 to 1.97 m/s. The results from the 3D model

(scenario SBL5) predict a slightly lower increase in the range of around 30 per cent, with recorded peak velocities of and 1.84 m/s during the ebb and flood tide generation respectively.

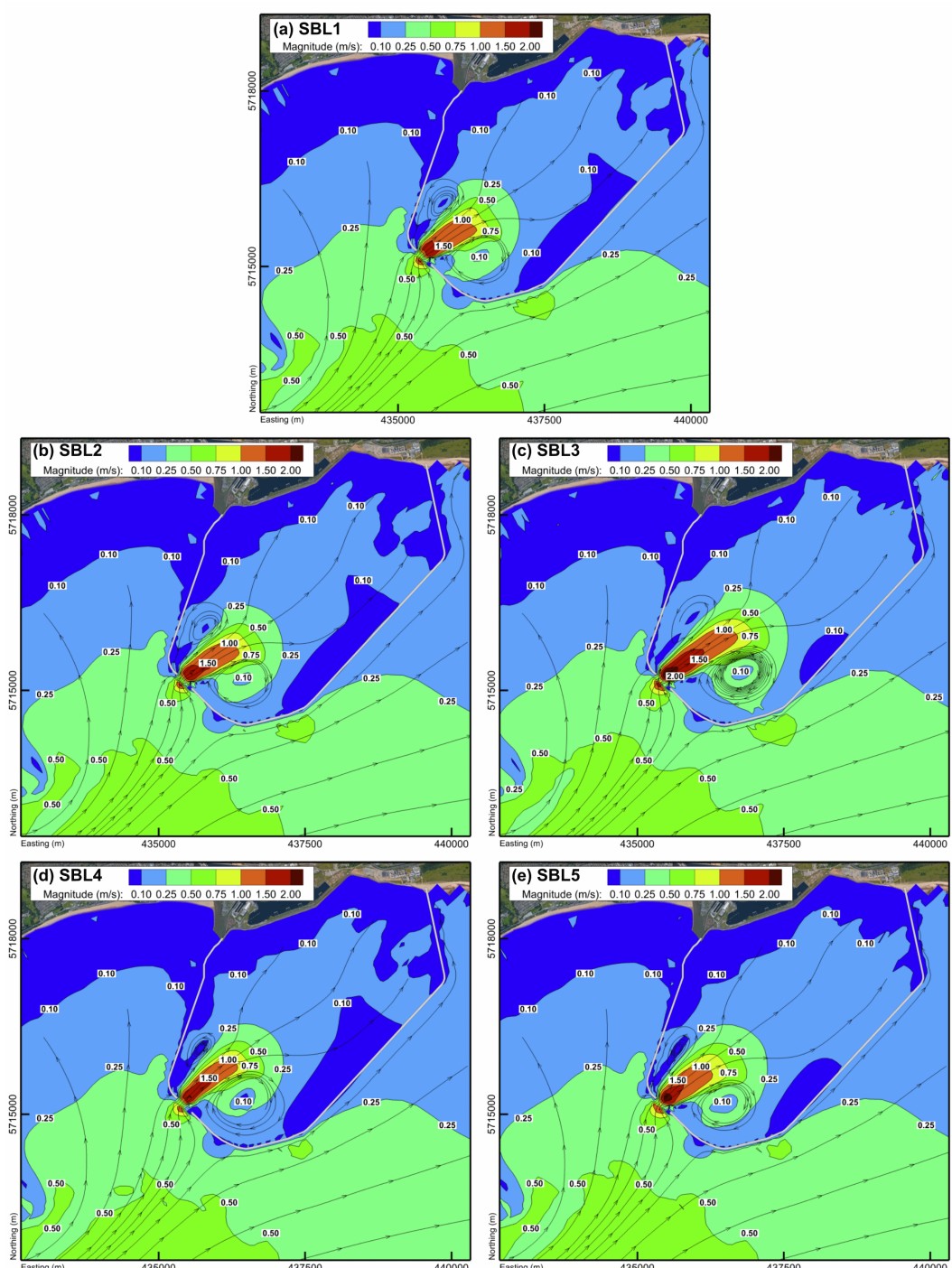

**Figure 12.** A comparison of velocity fields during the flood generation phase for a typical spring tidal cycle. (**a**) 2D simulation with no refinements (SBL1); (**b**) 2D with $A_{T1}$ = 216.1 m$^2$ (SBL2); (**c**) 2D with $A_{T2}$ = 38.5 m$^2$ (SBL3); (**d**) 3D with $A_{T1}$ = 216.1 m$^2$ (SBL4); (**e**) 2D with $A_{T2}$ = 38.5 m$^2$ (SBL5).

The 3D complexity of the velocity profile in scenarios SBL4 and SBL5 is highlighted in Figure 14. The predicted velocity profiles have been recorded at 50 m increments from the turbine exit at four locations. At 50 and 100 m downstream of the turbine, the higher velocity in the middle layers is still clear. However, the jet completely dissipates by the distance of 150 m, where velocity profile transitions to a conventional logarithmic shape of a free surface flow. This observation is consistent

with the findings of Jeffcoate et al. [22]. Even though the effects of the jet were lost after this distance, a direct comparison between the two scenarios indicated a clear difference between simplified and realistic velocity distribution. Scenario SBL5 still retains a higher depth-averaged velocity compared to SBL4. An evident increase in the flow velocity could be observed in all cases with the refined momentum treatment.

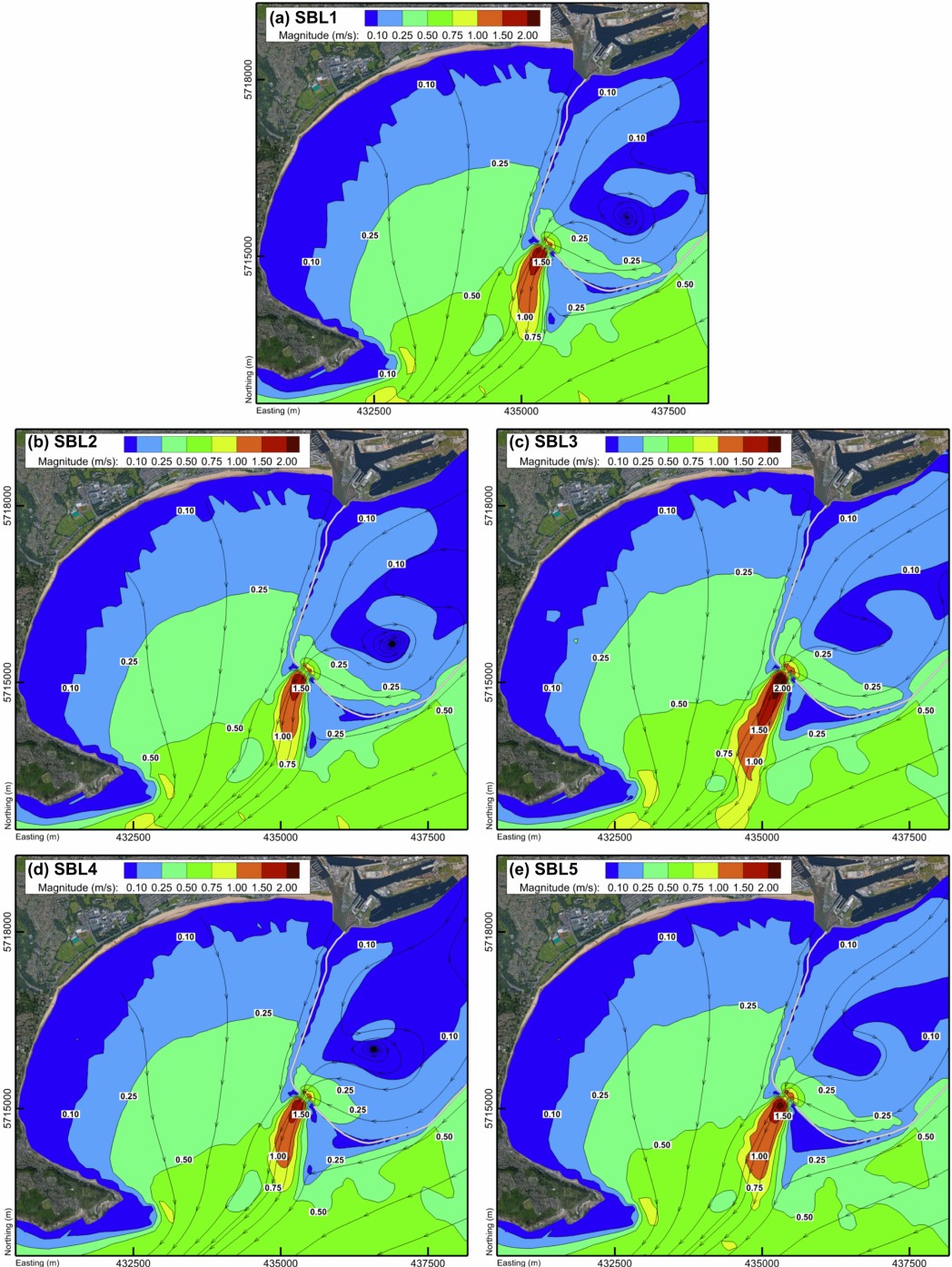

**Figure 13.** A comparison of velocity fields during the ebb generation phase for a typical spring tidal cycle. (**a**) 2D simulation with no refinements (SBL1); (**b**) 2D with $A_{T1}$ = 216.1 m$^2$ (SBL2); (**c**) 2D with $A_{T2}$ = 38.5 m$^2$ (SBL3); (**d**) 3D with $A_{T1}$ = 216.1 m$^2$ (SBL4); (**e**) 2D with $A_{T2}$ = 38.5 m$^2$ (SBL5).

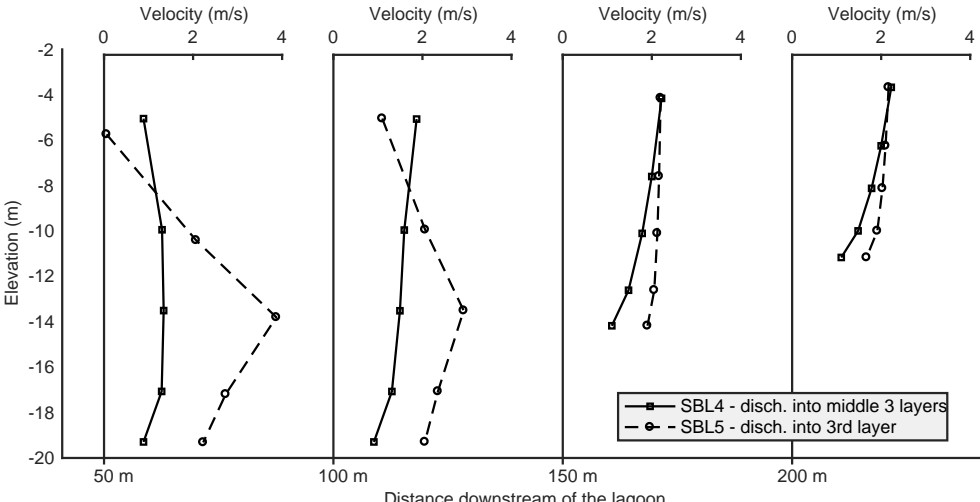

**Figure 14.** Representative 3D velocity profiles for SBL4 and SBL5 during the ebb generation in four locations downstream of the turbine during generation phase.

### 4.2. Water Levels

A comparison between the model predictions for the different scenarios highlighted that the treatment of the momentum had little influence on water surface elevations outside of the lagoon, where both high and low tide water levels were practically unaffected throughout the domain. However, there was a noticeable difference in the water levels inside the lagoon as illustrated in Figure 15 and Table 4. The figure shows the water level time series for a typical tidal cycle during a spring tide, with the same trend being observed for the complete 14-day neap-spring tidal cycle.

**Table 4.** Maximum and minimum water level values in meters recorded during a spring tide.

| Scenario | Max out (m) | % from SBL1 | Max in (m) | % from SBL1 | Min out (m) | % from SBL1 | Min in (m) | % from SBL1 |
|---|---|---|---|---|---|---|---|---|
| SBL1 | 4.24 | - | 2.80 | - | −4.05 | - | −2.72 | - |
| SBL2 | 4.24 | 0.0% | 2.83 | 1.1% | −4.05 | 0.0% | −2.72 | −0.8% |
| SBL3 | 4.25 | 0.1% | 3.24 | 16.0% | −4.05 | 0.1% | −2.72 | −16.8% |
| SBL4 | 4.22 | −0.6% | 2.80 | 0.2% | −4.04 | 0.1% | −2.72 | −0.4% |
| SBL5 | 4.22 | −0.5% | 3.09 | 10.6% | −4.05 | -0.2% | −2.72 | −11.6% |

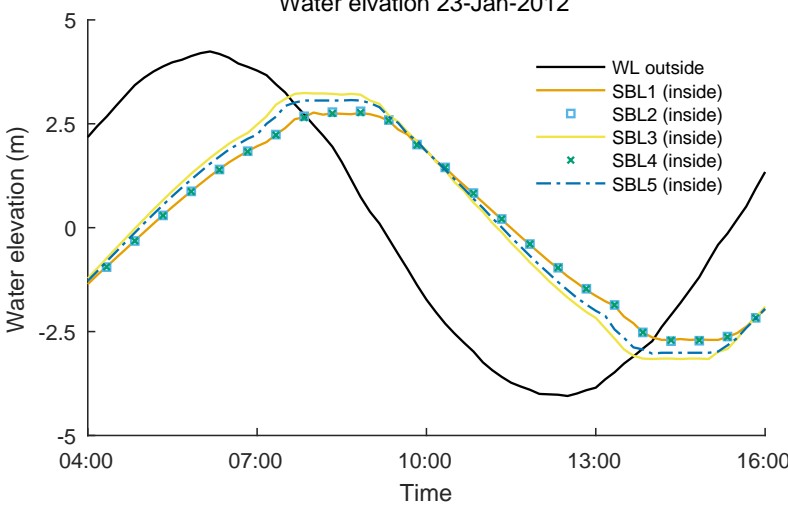

**Figure 15.** Comparison of water elevations inside the lagoon for the five different cases.

*4.3. Power Output*

Predicted annual energy potential is a critical factor when considering the construction of a tidal power plant. The instantaneous power production was calculated at each time step using the power-head relationship (Figure 9) and then integrated over the time interval of the simulation to obtain total electricity generated in that period. The simulation time was carefully selected to cover a typical neap-spring tidal cycle spanning over 14 days and taking into account the neap-spring variation in power production. Neglecting any tidal variations that could develop over the rest of the year, this neap-spring tidal cycle was presumed to be sufficient for this assessment. The estimated annual (neap-spring averaged) energy was then calculated by extrapolating the simulation results over the period of one year [15]. The total electricity generated for the baseline scenario was estimated to be about 578 GWh/year, which was consistent with predictions for past studies [2,13,49].

**Table 5.** Typical annual energy prediction extrapolated from a characteristic neap-spring tidal cycle for the five different cases.

| Scenario | Annual Energy (GWh) | % from SBL1 |
|---|---|---|
| SBL1 | 578.39 | - |
| SBL2 | 584.98 | 1.1% |
| SBL3 | 515.36 | −10.9% |
| SBL4 | 578.63 | 0.0% |
| SBL5 | 532.85 | −7.9% |

Unlike the local hydrodynamic conditions, the annual energy potential, which is mainly governed by the water level difference across the lagoon, was less sensitive to including momentum conservation in the numerical model. Predictions from all five simulations are summarised in Table 5. The values range from 515 GWh/year to 585 GWh/year. In scenarios SBL2 and SBL4, where a wider distribution of the velocity was predicted, the impact on the estimated power output was practically negligible with 1.1 per cent and less than 0.05 per cent, respectively. The predicted power output for scenarios SBL3 and SBL5, with the more realistic implementation of the jet, has decreased by 10.9 per cent and 7.9 per cent respectively. This is almost counter-intuitive, since we would expect a higher output due to the higher water level amplitudes observed inside the lagoon (Figure 15). However, this resulted in the lagoon operating at higher heads and generating over shorter time intervals, which considerably decreased its efficiency (Figure 9 and Equation (6)). Although including new and more accurate water levels in the optimisation of the lagoon would increase the efficiency of electricity generation, this re-emphasises the importance of accurate modelling of the scheme.

## 5. Discussion

We have described the background and motivation for the investigation of the momentum flux through hydraulic structures and particularly the tidal turbines. As highlighted by other researchers, an inaccurate representation of the simulated hydrodynamics through the turbines in a TRS can have a significant impact on predicting the performance and impacts of a tidal range scheme [1,2,23]. Tidal lagoons might be considered as relatively small structures with a lesser overall impact on the hydro–environment compared to larger schemes, such as barrages. However, due to a higher complexity of the high-speed flows exiting through a narrow section of the lagoon, a correct treatment of momentum conservation must be considered for higher accuracy of the predictions. This study provides a detailed methodology that has been developed for the refinement of momentum conservation and its application to both two- and three-dimensional hydrodynamic models for the assessment of tidal energy schemes. The model was employed for the proposed tidal lagoon in Swansea Bay, UK. After the initial validation of the hydrodynamic model, the model was run for five different

scenarios, covering both 2D and 3D simulations. Two different approaches to the distribution of the velocity at the turbine exit were investigated. The study analysed their effects on the hydrodynamic results as well as on the predictions of the annual energy output.

The method for momentum conservation used in this study introduces an additional term to the momentum equations, representing an equivalent external force acting on the volume of the computational cell across the turbine. The calculation of the force is highly sensitive to the assumption of the outlet velocity distribution and choosing the most realistic velocity profile is of utmost significance. As was demonstrated for scenarios SBL3 and SBL5, a simplified velocity distribution concentrated over a small flow-through, will result in a highly sophisticated 3D velocity field locally and will have an impact on both the water levels inside the lagoon and the power generation. These scenarios have produced visibly longer jets with high velocities reaching further away from the turbines before dissipating. However, a more realistic distribution of the flow velocity coming out of a diffuser (SBL2 and SBL4) better matched the 2D nature of the tidal hydrodynamics and showed less impact on the hydrodynamics compared to the simplified approach. The accuracy of the simulation results was particularly improved using the 3D model (SBL4) in the vicinity of the turbines, where an accurate vertical velocity profile of the jet was replicated by the model. The results showed that the simplified version gave worse results than not including the momentum at all. Suggesting that in situations where actual vertical velocity profile of the discharge is unknown a model with no momentum conservation will provide more accurate results than a model with an erroneously assumed vertical velocity profile. The realistic distribution gave very similar results to not including the momentum transfer, especially in the 2D case (SBL2). This suggests that the extra complexity of this technique and the use of a 3D model (SBL4) is only necessary for studies that are focused on the near-field effects of tidal range structures. For example, such approach will have significant advantage for studies of scour and deposition of sediment in the vicinity of the lagoon due to its operation. However, for regional-scale studies, where such local details are insignificant, the results showed the continuing scope for using 2D depth-average models without including the momentum conservation.

Regardless of the assumption for the velocity distribution, it was observed that 2D models generally predict a larger water level amplitude inside the lagoon compared to 3D models. The increased water levels in 3D models can be attributed to head losses induced by the jets, which could not be captured by the 2D model. Similar observations were made by Jeffcoate et al. [22], through an experimental study which showed that depth-averaging fails to capture the higher complexities of the flow field exiting the turbines. The effects of the correct momentum transfer on the hydrodynamics were constrained to a relatively local area. As observed in Figure 13, there was very little change in the hydrodynamics at a distance of about 20 times the turbine diameter downstream of the turbines. However, this could have a significant impact on solute transport modelling and sediment fluxes, particularly in the vicinity of the turbines. Models at the regional scale also fail to include the rotational velocity of the flow exiting the turbines. To include such level of detail, it would be necessary to model the hydrodynamics at a much higher grid resolution, which is unnecessary for simulating the hydrodynamics characteristics at the regional scale.

Future work could consider increasing the number of layers of the 3D model. For example, a study by Lin et al. [46] employed ten computational layers to model adequately the wake behind a tidal stream farm. However, each additional horizontal layer adds to the total computational time and makes the regional model less attractive for practical use and with limitations on the resources available for only a slight increase in the accuracy of the predictions. A multi-scale approach could prove to be more appropriate, where the high-resolution sub-grid scale model can be dynamically linked to a local or regional scale model. This could prove particularly useful for the investigation of scour and sediment transport associated with the turbine operation. The significance of representing the momentum flux accurately through hydraulic structures, as outlined in this study, highlights the importance of accurately predicting the jet characteristics, particularly in the assessment of hydro–environmental impacts of such structures.

## 6. Conclusions

The paper presents and discusses a novel approach for modelling the momentum flux through hydraulic structures, such as tidal range turbines, for regional-scale hydrodynamic models. The theoretical background and key formulations of the methodology have been discussed, as well as its implementation into the Delft3D numerical model. The proposed Swansea Bay tidal lagoon was selected as a case study for assessing the effects of the introduced methodology on the hydrodynamic results. Furthermore, the study assessed the advantages and disadvantages of a 3D hydrostatic model over a 2D shallow-water model when predicting the hydro–environmental impacts of tidal range structures.

The results obtained indicated that modelling of marine structures with a high-velocity discharge (e.g., a jet discharged from a turbine within a tidal range power plant) should employ momentum conservation methods, particularly when assessing near-field hydro–environmental impacts. The effects of momentum conservation were constrained to the area in the vicinity of the lagoon and had no significant impacts on the far-field conditions. Including the momentum conservation has, however, enlarged both the length and magnitude of the jets. Careful consideration has to be given to the assumption of the discharged velocity distribution. The comparison between realistic (dissipated velocity flow through diffuser) and simplified (ignoring diffuser) distributions showed that the latter had a bigger influence on the hydrodynamics and power output estimation. Using a 3D model has also increased the accuracy of the predicted velocity field, particularly in terms of the vertical velocity distribution. However, these 3D effects were again limited to the vicinity of the hydraulic structures, with no significant advantage being shown in using a 3D model beyond the point where jets have been fully dissipated in a macro-tidal basin.

In general, the 3D hydrodynamic numerical model, including accurate representation of the momentum source through the turbines, has significantly improved the hydrodynamic results within the model domain. While a simplified 2D model might be sufficient for assessment of regional-scale impacts, the 3D model with the full momentum conservation method proved to be essential for accurate assessment of local conditions. For instance, assessment of the velocity distribution and magnitude of the discharged jet, are directly associated with hydro–environmental impacts, such as increased erosion and changes to the sediment transport patterns.

**Author Contributions:** N.Č. was the Research Associate on the project and set-up the model applications and undertook the studies, including the development of the methodology for momentum conservation and its implementation into the Delft3D software. R.A. and R.A.F. have been leading a wide range of research projects in tidal range modelling and were instrumental in developing the methodology for momentum conservation. N.Č. drafted the manuscript, which was refined by all authors.

**Funding:** This research was funded by the Engineering and Physical Sciences Research Council (EPSRC) grant number EP/L016214/1.

**Acknowledgments:** The study was part of a PhD research topic within the Centre for Doctoral Training in Water Informatics: Science and Engineering (WISE CDT). Additionally, we acknowledge the support through the INTERREG Smart Coasts project (Ireland Wales Cross-Border Programme 2007-2013 (INTERREG 4A) project 065: Smart Coasts = Sustainable Communities SCSC') and Professor David Kay for providing the field data used in the research.

**Conflicts of Interest:** The authors declare no conflict of interest. The funders had no role in the design of the study; in the collection, analyses, or interpretation of data; in the writing of the manuscript, or in the decision to publish the results.

## Abbreviations

The following abbreviations are used in this manuscript:

| | |
|---|---|
| TRS | Tidal range structures |
| CFD | Computational fluid dynamics |
| RANS | Reynolds-averaged Navier–Stokes equations |
| DCO | Development consent order |

ADI     Alternating direction implicit method
CD      Chart datum
MSL     Mean sea level
ADCP    Acoustic Doppler current profiler
NSE     Nash-Sutcliffe model efficiency

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
