# Peer review of "Implementation of a Full Momentum Conservative Approach in Modelling Flow Through Tidal Structures"

_water, doi:10.3390/w11091917_

Round 1
Reviewer 1 Report
The paper is well written. I have only a few concerns and suggestions.
A zoomed image of the lagoon with all turbine locations would be helpful to the readers. Why or how are the locations of sites L1, L2, etc chosen? Boundary conditions are crucial, specially for the lagoon and turbines. A short overview of the model setup and boundary conditions should be presented before the 'model validation' section. The paper details the modeling approach later, which is fine. Is the Manning's roughness coefficient constant across the domain? The line 153 seems to suggest that. What is the rationale? A calibration is mentioned, but no details are presented.Overall, it is a great paper. Are these results published for the first time?
Reviewer 2 Report
In this paper the potential impacts on the hydrodynamics provoked by the operation of the proposed Swansea Bay tidal lagoon are analysed. With this aim, a shallow water numerical model is analysed by considering the momentum conservation approach. The methodology implemented is appropriate considering the spatial scale of this study (CFD would be more appropriate for reduced scales). The results are of interest for analysing the impacts of this proposed structure and the same procedure could be applied for the analysis of other structures.
All in all, I recommend this paper for being accepted for publication in water. However, the following comments could be considered in the revision of the document prior its publication:
-A more detailed state-of-the-art could be provided. In particular, this momentum approach with DELFT model was previously used in the case of the Galician Rias for analysing the potential impacts of the operation of a tidal stream farm.
-It could be interesting to show if 5 layers are enough to model the turbine and the resulting impacts. Usually 3D hydrodynamic models are composed of 12 layers or more. However, it is likely that in this case 12 layers are not necessary (given the hydrodynamic characteristics of the area), but it should be appropriate to demonstrate that 5 are sufficient.
-At least a brief conclusions section seems to be appropriate (not discussion and conclusions)
